# Intake of Antioxidant Vitamins and Minerals in Relation to Body Composition, Skin Hydration and Lubrication in Young Women

**DOI:** 10.3390/antiox10071110

**Published:** 2021-07-12

**Authors:** Anna Puścion-Jakubik, Renata Markiewicz-Żukowska, Sylwia K. Naliwajko, Krystyna J. Gromkowska-Kępka, Justyna Moskwa, Monika Grabia, Anita Mielech, Joanna Bielecka, Elżbieta Karpińska, Konrad Mielcarek, Patryk Nowakowski, Katarzyna Socha

**Affiliations:** Department of Bromatology, Faculty of Pharmacy with the Division of Laboratory Medicine, Medical University of Białystok, Mickiewicza 2D Street, 15-222 Białystok, Poland; renmar@poczta.onet.pl (R.M.-Ż.); sylwia.naliwajko@umb.edu.pl (S.K.N.); krystyna.gromkowska.kepka@gmail.com (K.J.G.-K.); justyna.moskwa@umb.edu.pl (J.M.); monika.grabia@umb.edu.pl (M.G.); anita.mielech@umb.edu.pl (A.M.); joanna.bielecka@umb.edu.pl (J.B.); elzbieta.karpinska@umb.edu.pl (E.K.); konrad.mielcarek@umb.edu.pl (K.M.); patryk.nowakowski@umb.edu.pl (P.N.); katarzyna.socha@umb.edu.pl (K.S.)

**Keywords:** vitamin A, vitamin C, vitamin D, vitamin E, copper, manganese, zinc, body composition, skin, antioxidants

## Abstract

The aim of this study was to estimate the consumption of selected dietary components with antioxidant properties, undertake body composition analysis, assess skin hydration and lubrication, and establish the relationships between the above parameters. The study was carried out on 172 young women. The consumption of ingredients (vitamins A, C, D and E, and Cu, Mn, Zn) was assessed using the Diet 6.0 program, body composition was assessed using electrical bioimpedance and skin hydration and lubrication were assessed using the corneometric and sebumetric methods, respectively. About one-third of students showed insufficient consumption of vitamin C, vitamin E and zinc, while about 99% showed insufficient vitamin D levels. The highest degree of hydration was observed in the areas of the eyelids, neckline and chin. The greatest amount of sebum was found in the area of the nose and forehead. Low positive correlations between hydration or lubrication and Cu, vitamin A and vitamin E were observed. In conclusion, to properly moisturize and lubricate the skin, young women should eat products that are rich in ingredients with antioxidant properties, in particular fat-soluble vitamins A and E, but also copper.

## 1. Introduction

The skin is a barrier that separates the human body from the external environment. It takes part in many processes, including metabolic and immune processes, and has a thermoregulatory and protective function against pathogenic microorganisms. This organ can fulfill its functions only when it is moistened and the hydro-lipid layer is properly composed. Skin condition is influenced by a number of factors, including genetic and environmental factors (diet, smoking and weather conditions, among others) [1,2]. Many of these factors can disrupt the oxidative/reductive status.

Both extracellular and intracellular oxidative stress, which are initiated by reactive oxygen species, can result in skin pigmentation disorders, as well as premature aging. Overexposure to ultraviolet (UV) radiation can accelerate this process [3].

A properly balanced diet should provide ingredients from all categories, namely, basic nutrients, vitamins and minerals, in accordance with the standards established by experts [4,5]. An important role in the pathogenesis of many diseases, including skin diseases, is played by the deficiency of ingredients with antioxidant properties [6,7,8,9].

Disturbed oxidative balance can cause premature skin aging but also lead to the development of diseases, such as acne, rosacea, skin cancer, psoriasis, vitiligo, scleroderma, contact dermatitis, lichen planus and chronic venous ulcers [10].

Vitamin A is involved, among others, in the protection against the effects of reactive oxygen species, in the proper functioning of the immune system, in cell division and differentiation and in maintaining their proper structure. It should be emphasized that this vitamin contributes to the normal maintenance condition of the epidermis, inter alia, by regulating the process of exchange and exfoliation of the outer layers of cells. The external application of retinol may contribute to the reduction of wrinkles. Moreover, retinol helps to increase the accumulation of hyaluronic acid in the epidermis, which plays an important role in moisturizing the skin. Retinoids inhibit transepidermal water loss (TEWL), improve the protective function of the epidermis, prevent collagen degradation and can be a chemopreventive agent [11,12,13,14].

Vitamin C is water soluble, participates in collagen biosynthesis and helps in antioxidant protection in the case of photodamage caused by UV radiation. In addition, it prevents scurvy and degenerative diseases [15,16]. Moreover, it affects the absorption of calcium and iron consumed with meals or pharmaceutical preparations. Additionally, vitamin C helps to seal the capillaries [4,17].

Vitamin D is primarily involved in calcium phosphate metabolism, for example, bone metabolism. Studies have shown the presence of vitamin D receptors in extra-skeletal organs, including the brain, heart and intestines. The pleiotropic effect of vitamin D was tested in numerous studies and the results suggest a relationship between its low concentration in blood serum and an increased risk of cancer, immune-related diseases, immune disorders, civilization diseases, psychiatric disorders and neurodegenerative diseases [18,19,20]. Furthermore, research confirms its diverse functions and role in the etiology of various dermatological diseases, such as acne, atopic dermatitis and melanoma [21,22].

Vitamin E has an anti-aging effect. This vitamin has the ability to neutralize free radicals in a hydrophobic environment. Literature data indicate that it may protect against the risk of developing, for example, atherosclerotic lesions, coronary artery disease [23,24], cataracts and cancers [25]. It is used in the form of both oral supplementation and topical therapy due to its photoprotective and skin-barrier-stabilizing properties [26].

Copper (Cu) is a part of superoxide dismutase, the main enzyme involved in the breakdown of free radicals. It takes part in the formation of bonds in collagen and elastin, as well as in the synthesis of melanin, which is the dye of hair and skin [27].

Manganese (Mn) activates enzymes that are involved in the synthesis of proteins, fatty acids and nucleic acids, and also participates in the metabolism of thyroid hormones. It is included in, among others, superoxide dismutase (SOD), protecting the body against free radicals. It is necessary for the proper formation of connective tissue and the appearance of the skin [28].

Zinc (Zn) has regulatory, structural and catalytic functions, and it is a component of over 300 enzymes, for example, Cu/Zn-SOD. This microelement participates in the metabolism of essential nutrients, such as proteins, fats and carbohydrates, as well as in energy metabolism and the functioning of many hormones. The skin has the third-highest zinc content in the human body. Zinc deficiency is associated with diseases such as pellagra, delayed wound healing and alopecia. In addition, it regulates the secretion of sebum, has anti-inflammatory and anti-blackhead properties and is used in the treatment of acne [4,29].

The above dietary ingredients with antioxidant properties have a positive effect on the processes taking place in the skin. The aim of the study was to assess the consumption of ingredients with antioxidant properties (vitamins A, C, D and E, and Cu, Mn and Zn), body composition and hydration and lubrication of the skin. Moreover, this study aimed to determine whether the dietary intake of antioxidants or body composition correlated with skin condition. As far as we know, it is the first publication of this type that combines the above parameters.

## 2. Materials and Methods

### 2.1. Study Group

The study was conducted among young women (*n* = 172) who were students of laboratory medicine, pharmacy, cosmetology and dietetics at the Medical University of Białystok, Poland. All participates were residents of cities with more than 300,000 inhabitants. The women were between 18 and 25 years old, their heights ranged from 155 to 182 cm and their body weights ranged from only 39 to 100 kg (Table 1). All study participants declared that they did not smoke cigarettes.

All the students agreed to perform the analyses. The study was approved by the Bioethics Committee of the Medical University of Białystok (R-I-002/8/2015 and R-I-002/39/2015).

### 2.2. Estimating the Consumption of Antioxidant Ingredients

Three 24 h nutritional interviews were conducted with each student using the current record method, namely, basic on ‘Album of photographs of food products and dishes’ [30]; the interviews included one free day and 2 working days. 

Based on the nutritional interviews, the intake of vitamins and minerals with antioxidant properties was calculated using the Diet 6.0 computer program. This program was developed by the National Food and Nutrition Institute in 2018 [4,31,32]. It includes over 2100 products and dishes, and approximately 1300 dietary supplements.

The obtained results were compared with the currently applicable standards at the levels of: estimated average requirement (EAR) and adequate intake (AI). Factors such as age, gender and physiological condition were taken into account. The results were compared to the norm at the EAR level in the case of: vitamin A (500 μg retinol equivalent (RE)), vitamin C (60 mg), Cu (0.7 mg) and Zn (6.8 mg), and at the level of AI in the case of vitamin D (15 µg of cholecalciferol), vitamin E (8 mg of α-tocopherol equivalent) and Mn (1.8 mg) [4]. Moreover, the percentage of people with insufficient and sufficient consumption was calculated.

EFSA recommendation requirements differ from national or regional standards: vitamin A (average requirement (AR): 570 μg RE), vitamin C (AR: 90 mg), vitamin E (AI: 13 mg of α-tocopherol equivalent), Cu (AI: 1.3 mg), Mn (AI: 3.0 mg) and Zn (AR: 6.2 mg). In the case of vitamin D, the recommendation is the same (AI: 15 µg of cholecalciferol) [5].

### 2.3. Body Composition Analysis

Body composition was assessed using the InBody 720 (Biospace, Eonju-ro, Korea) device, which is based on the bioelectrical impedance analysis (BIA) method. It is medical equipment with high accuracy. The principle of the method is based on measuring the body’s resistance. The human body is divided into 5 cylindrical parts, which are fed with currents of different voltages. The device uses 8-point electrodes to improve the accuracy of the measurement. This method is safe, non-invasive and reliable. 

During the study, the following parameters were measured: indicative of physical activity (fitness score), circuits (abdomen circumference, arm muscle circumference, chest circumference, hip circumference), indicating the content of bone minerals bone mineral content), concerning the basal metabolism (basal metabolic rate), the content of muscle and fat tissue (body fat mass, fat-free mass, mineral mass, obesity degree, percent body fat, protein mass, skeletal lean mass, skeletal muscle mass, visceral fat area) and water content in different areas (extracellular fluid/total body fluid: ECF/TBF, extracellular water/total body water: ECW/TBW, extracellular water mass, intracellular water mass, total body water mass).

### 2.4. Skin Preparation and Measurement Conditions

Before starting the study, the students rested for 10–20 min in order to stabilize blood circulation and reduce the impact of physical activity on skin hydration and lubrication. The areas of skin (cheeks, chin, eyelids, forearm, forehead, neckline, nose) on which the measurements were made were previously cleaned with a make-up remover and then with water at home on the day of the analysis.

The measurements were performed in a room with a temperature of about 20 °C and air humidity ranging from 40 to 60%.

### 2.5. Measurement of Skin Hydration

Skin moisture was measured with a Corneometer CM 825 (Courage + Khazaka Electronic, Köln, Germany). This is a capacitive method. The measurement is based on the difference between the dielectric constant of a substance (usually below 7) and water (81). Depending on the changing water content, the measuring capacitor shows changes in capacitance. There are 2 tracks in the probe head. An electric field is created between them. One path has an electron deficiency (positive sign) and the other path has an electron excess (negative sign). When measuring skin moisture, the scattering field penetrates the first layer (10–20 μm of the stratum corneum) of the skin and the capacitance is determined on this basis. Three measurements were taken in each area and the result is presented as an average.

A score below 30 units (u.) pointed to very dry skin, from 30 to 40 meant that the skin was dry and above 40 meant that the skin was determined as sufficiently moisturized [33].

### 2.6. Measurement of Skin Lubrication

The skin lubrication was measured with the Sebumeter SM 815 (Courage + Khazaka Electronic, Köln, Germany). It is a direct, photometric method that is based on the measurement of sebum secreted on the skin. A measuring tape with an area of 64 mm^2^ and placed in the cassette was applied to the skin for 30 s. The measuring head of the cassette was placed in the aperture of the device and the photocell measured the transparency of the tape. In the studied areas, 3 measurements were made and the final result is the average of these readings. 

The obtained result ranged from 0 to 350 µg/cm^2^. Depending on the examined area, the following criteria were adopted for the degree of skin lubrication: dry skin (reading <100 µg/cm^2^ on the forehead and in the T zone, <70 µg/cm^2^ on the cheeks), normal (from 100 to 200 µg/cm^2^ on the forehead and in the T zone, from 70 to 180 µg/cm^2^ on the cheeks), oily (over 220 µg/cm^2^ on the forehead and in the T zone, over 180 µg/cm^2^ on the cheeks) [33].

### 2.7. Statistical Analysis

The obtained numerical data were analyzed using Microsoft Office Excel 2019 and Statistica 13.3 Software (StatSoft, Tibco, Palo-Alto, CA, USA). The normality of the data distribution was assessed using the Kolmogorov–Smirnov, Lilliefors and Shapiro–Wilk tests. Descriptive statistics parameters were also calculated: mean, standard deviation, median, minimum, maximum and lower and upper quartiles.

The Kruskal–Wallis ANOVA test was used for the statistical analysis. Spearman’s rank-order correlation coefficients were also determined. The level considered as statistically significant was *p* < 0.05.

## 3. Results

It was shown that students were mainly characterized by insufficient intake of vitamin C (39.5% of the students) and zinc (32.0%). Sufficient intake of vitamin D was found in only 1.2% and sufficient vitamin E in 26.7% (Table 2).

Table 3 presents the descriptive statistics for the female body composition analyses. The average fitness score was 72.77 ± 5.08 points, which indicates that this group was moderately active. Moreover, it was shown that the intracellular water mass was on average 20.09 ± 2.33 L and extracellular water mass was 12.32 ± 1.44 L. It was shown that the mean value of the total body water mass parameter was 32.40 ± 3.75 kg. Importantly, it was also noted that the ECF/TBF averaged 0.333 ± 0.004 and the ECW/TBW averaged 0.380 ± 0.005.

As part of this study, the hydration of selected areas of the body and face was assessed. It was shown (Figure 1) that individual areas differed significantly in terms of their water content in the epidermis (*** *p* < 0.001). The eyelids (67.02 ± 11.35 u.), neckline (62.14 ± 10.41 u.) and chin (59.66 ± 10.90 u.) had the highest mean water contents in the epidermis, and the nose (26.55 ± 16.21 u.) had the lowest water content.

The assessment of the lubrication of the face skin showed that the most sebum was present on the skin of the nose (105.13 ± 73.53 µg/cm^2^) and forehead (103.62 ± 65.69 µg/cm^2^). The degree of lubrication significantly depended on the area of the skin (Figure 2).

In the next step, an assessment of what percentages of the female students were characterized by each of the three levels of skin hydration—very dry, dry and sufficiently moisturized—in individual areas took place. It was shown that the highest percentage of female students had sufficient hydration of the eyelids (98.2%), neckline (97.6%) and chin (96.0%). The nose was the area that had very dry skin for the largest group of female students (59.3%) (Table 4).

When assessing the lubrication, the results were classified into three categories: dry, normal and oily skin. It was shown that the highest percentage of female students was characterized by dry skin on the chin (64.5%) (Table 5).

It was shown that in the case of skin hydration, the vast majority of female students had average skin hydration, defined as sufficient (*n* = 166), and only *n* = 6 female students had dry skin.

Different results were obtained in the case of skin lubrication. It was shown that *n* = 102 female students had dry skin, *n* = 69 women had normal skin, and only *n* = 1 had oily skin. Therefore, for the analysis, the purpose of which was to show differences in the consumption of ingredients with antioxidant properties, two groups were selected: people with dry (*n* = 102) and normal skin (*n* = 69). Interestingly, in the case of women with normal skin, higher average consumption of all tested ingredients was noted, but in no case were these differences statistically significant (Figure 3).

The assessment of the correlation between the consumption of the analyzed vitamins and minerals with antioxidant properties showed significant correlations between almost all diet components (Appendix A). The highest relationships were noted between the consumption of Cu and Zn (*r* = 0.74), Cu and Mn (*r* = 0.66) and Mn and Zn (*r* = 0.62).

Relationships between the hydration and lubrication of examined skin areas were also searched for (Appendix A). There were no correlations between total hydration and total lubrication, only correlations between single areas. For example, there was a moderate positive correlation between the hydration of the forearm and neckline.

Additionally, the relationship between skin hydration and lubrication and the results of body composition analysis were assessed in the group of female students (*n* = 172). There was a negative correlation between age and cheek skin hydration (*r* = −0.19), as well as between body moisturizing (forearm and neckline skin) and ECW/TBW (*r* = −0.15). Positive correlations occurred in the case of the lubrication of skin and ECF/TBF (*r* = 0.15) (Table 6).

The relationship between skin hydration and lubrication and the consumption of selected ingredients with antioxidant properties was most evident in the case of vitamins A and E, as well as Cu. There was a low positive correlation between total skin hydration and vitamin A consumption (*r* = 0.17) and a low positive correlation between skin lubrication and vitamin E consumption (*r* = 0.15) and Cu consumption (*r* = 0.17) (Table 7).

## 4. Discussion

This study focused on the search for relationships between skin hydration and lubrication, body composition and the consumption of ingredients with antioxidant properties. Lubrication and hydration of the skin can be directly related to nutritional status and body composition analysis results are indirectly related to eating habits. Literature data show a strong relationship between nutrition and skin condition, but little research has focused on the relationships investigated in this publication.

Properly moisturized skin creates a barrier that protects organs and tissues against mechanical, chemical and biological factors [34]. There are four basic types of skin in the literature: dry, normal, oily and combination [2].

Dry skin is a type of skin with a high degree of epidermal dehydration. It is characterized by an abnormal process of keratinization and exfoliation of epidermal cells and impaired lipid production, which contributes to the formation of dryness [35]. Our research showed that 102 female students had dry skin.

Normal skin is rarely found in young women. This type of skin is characterized by smoothness and firmness, as well as the absence of skin defects [36]. According to the results of our study, 69 young women had normal skin.

Oily skin is characterized by excessive activity of the holocrine sebaceous glands. This type usually occurs in young people and during adulthood. Excess sebum on the surface of the skin and keratinization disorders can lead, among others, to the development of acne [37]. This type of skin was identified in only one person.

Combination skin is the most popular type of skin and is a combination of dry and oily skin. The dry U zone is distinguished (around the eyes and temples) and the oily T-zone (forehead, nose, cheeks, chin). This type is characterized by disturbed hydration on the epidermis surface [37].

Each type of skin requires carefully selected care, as well as a diet that is aimed at reducing defects and possibly supporting the treatment of skin diseases.

We showed that the areas of the body and face of the young women differed significantly in terms of skin hydration, and the areas that were best moisturized for the majority of them were the eyelids (98.2% of women), neckline (97.6% of women), chin (95.9% of women) and forehead (88.4%) (Table 4).

In the case of skin lubrication, we also noted statistical differences between the examined skin areas. Normal skin was recorded on the cheeks in 39.5% of the examined women and on the forehead in 37.8% of the women (Table 5).

As part of this study, the impact of the consumption of selected dietary components with antioxidant properties was assessed: vitamins A, C, D and E, as well as Cu, Mn and Zn.

Vitamin A is one of the basic antioxidant vitamins. As shown in the present study, the average daily consumption was 825 ± 688 μg retinol equivalent (RE) and 27.3% of young women were characterized by a low dietary intake of this vitamin (Table 2). In the case of deficiency, the diet can be enriched with products such as: meat, butter, dairy products and eggs, as well as vegetables and fruits (carrots, red peppers, dark green leafy vegetables, melons and mangoes) [38]. The EFSA report summarizes data gathered from national reports about the intake of nutrients in European countries. According to the EFSA average, vitamin A intake ranged between 816 and 1498 μg RE/day in adults [5]. Gacek [39] conducted a study among 120 women aged 19–25 who regularly engaged in physical activity. The supply of vitamin A was at the level of 706.8 ± 111.2 μg/day (117.8% of the norm). The research carried out in our study showed that both women with dry skin and women with normal skin had a higher intake of vitamin A (777.03 vs. 901.66 μg/day, respectively). A study conducted in a group of 1004 young women aged 20–34 showed that the average consumption of vitamin A was as much as 1017 μg, which was a value that was higher by 240 μg/day than among the women in our study that were characterized by dry skin [40]. Research conducted by Gogojewicz et al. [41] on women aged 20–40 showed that the average consumption of vitamin A in the group of women practicing fitness (*n* = 20) was 488.1 ± 336.1 μg (the norm coverage was at the 69.7% level), while in the control group: 719.3 ± 963.8 μg (102.7% of the norm).

The analysis of consumption showed that the average dietary intake of vitamin C reached 82.6 ± 54.1 mg/day and almost 40% of the women consumed too little vitamin C. Our results are similar with data obtained in other European countries, where the average vitamin C intakes range from 65 to 138 mg/day in women [5]. A study conducted on young women [39] showed that the average vitamin C intake was 78.5 ± 15.2 mg/day (130.9% of the norm). This level was similar to the level of consumption by students from our research that had dry skin. Other authors estimated for the group of young women (*n* = 1004) that the average consumption of vitamin C was 79.1 mg/day, which was a value similar to the consumption of vitamin C by young women with dry skin in this present study [40]. In the group of young women practicing fitness (*n* = 20), as well as in the control group (*n* = 20), a low consumption of vitamin C was found at the levels of 32.9 ± 19.5 mg/day and 36.1 ± 21.6 mg/day, respectively, which accounted for 44.0% and 48.0% of the implementation of the standards [41]. These values were much lower than those shown in our study. It is commonly believed that citrus fruits and in their juices (lemon, oranges, grapefruits, bergamot) are products that are rich in vitamin C; however, much larger amounts can be found in blackcurrant, chokeberry, tomatoes, green and red peppers, strawberries, kiwifruit and green leafy vegetables, such as broccoli [42,43].

In the case of vitamin E, the average daily dietary intake (6.9 ± 4.1 mg) was lower than AI (8 mg) and only 26.7% of students ingested enough of it. Due to this fact, the intake of the following products should be increased: vegetable oils (sunflower, safflower) and nuts (almonds, hazelnuts) [25,44]. According to the EFSA report [5], in other countries, the average α-tocopherol intake was higher and ranged between 7.8 and 12.5 mg/day in women. The average consumption of vitamin E in local studies was at the level of 7.7 ± 1.4 mg/day (98.6% of the norm), which was a value higher than the consumption in both groups studied by us [39]. The other study on young women from Poland (*n* = 1004) showed that the average daily vitamin E intake was higher (9.4 mg) [40] than in our study. The consumption of vitamin E in a group of 20 young women from Poland doing fitness exercises was shown to be at the level of 4.1 ± 1.3 mg/day, while in the control group, it was at the level of 5.4 ± 2.0 mg/day, which were 51.2 and 67.8% of the standard, respectively [41]. The average consumption was lower than that of the female group in our study.

Only 1.2% of the surveyed women ingested a sufficient amount of vitamin D with the diet. It would be recommended to enrich the diet with products that are good sources of vitamin D, such as fish (tilapia, salmon and herring) [45]. Research confirms that fish consumption (300–1000 g per week) increases the concentration of vitamin D in serum [46]. However, the main factor responsible for the vitamin D content in the body is cutaneous synthesis [47]. Therefore, vitamin D intake is rarely assessed. For example, studies conducted in a group of 161 women showed that the average vitamin D intake was at the level of 2.45–2.92 µg/day, meaning that the percentage of compliance with the norm was 49% [48]. Our research showed a much larger anomaly.

In our research, we found that the consumption of Cu, Mn and Zn was 1.0 ± 0.4, 3.9 ± 1.9, 8.1 ± 2.3 mg/day, respectively. Insufficient intake was the most severe for Zn consumption (32%). Our results are similar to the EFSA report [5], where the average Cu intakes ranged between 1.15 and 2.07 mg/day, the mean Mn intakes of adults generally ranged around 3 mg/day (from 2 to 6 mg/day) and the average Zn intake ranged from 8.0 to 14.0 mg/day in adults. Omeljaniuk et al. [49] showed that the average consumption of Cu at the level of 0.87 ± 0.3 mg/day, Mn: 3.48 ± 1.6 mg/day, and Zn: 7.58 ± 2.3 mg/day. The percentages of women with insufficient consumption were 18%, 18% and 45% for Cu, Mn and Zn, respectively. Another study conducted among 161 students showed the consumption of Cu at the level of 0.86 ± 0.31 mg/day (which was 95.2% of the norm), the consumption of Mn at the level of 3.61 ± 1.56 mg/day (200.5% of the norm) and the consumption of Zn was 7.25 ± 2.40 mg/day (90.5% of the norm) [48]. Our previous research showed an interesting relationship: physically active students were characterized by higher consumptions of, among others, Cu, Mn and Zn. In the group of students who did not exercise but who were physically active, the values were the following: in the case of Cu—1.09 ± 0.7 mg/day vs. 1.22 ± 0.6 mg/day), in the case of Mn—4.68 ± 2.3 mg/day vs. 5.09 ± 2.0 mg/day, while in the case of Zn—9.66 ± 4.9 mg/day vs. 12.47 ± 5.9 mg/day [50]. However, our research showed insufficient intake of Cu in 20.9% of the group. Too low Zn intake was reported in almost one-third of the young women. The most beneficial result for the consumption of ingredients with antioxidant properties was obtained for Mn. It was shown that 91.9% of the respondents ingested a sufficient amount of this ingredient with their diet.

Body composition analysis is a common, non-invasive method for assessing fat and lean body mass. Body composition analyzers differ, among other things, in the degree of advancement, the number of parameters tested and their accuracy. Haq et al. [51] assessed, inter alia, body composition in a group of students of medical universities in China. The study was conducted on 695 students, including 471 females. In the group of young women, the BMI was at the level of 21.6 ± 3.7 kg/m^2^, while the fat percentage was 28.5 ± 7.9%. The group of young women we studied (*n* = 172) had a similar BMI (21.9 ± 3.2 kg/m^2^) (Table 1) and percentage of body fat (27.39 ± 6.47%) (Table 3).

Contemporary studies assessed the relationship between body composition and the emergence of civilization diseases, as well as skin diseases. However, there are still few scientific reports on the influence of body composition on the hydration and lubrication of the skin. In our work, we tried to assess these dependencies.

The control of body weight and parameters determining body composition is important because excessive fat gain adversely affects the functioning of organs and well-being. Often it is also the basis for the development of diseases, such as type 2 diabetes, insulin resistance and hypertension, which consequently also affect the appearance of the skin.

We have shown that better skin hydration is associated with a lower index of edema. Proper hydration results in a sufficient amount of extracellular water, while when its supply is disturbed, excess fluid may accumulate in the intercellular space, causing swelling. The opposite tendency occurs in the case of the assessment of lubrication of skin, which is indicated by a positive correlation.

The corneometric and sebumetric methods that we used are the reference methods. In the literature, they are mainly used to: compare other methods to them [52,53], the impact of supplementation (e.g., collagen [54], evaluation of the effectiveness of new formulations and uses of ingredients [55,56,57] and the external use of preparations [58,59,60,61]). To our knowledge, this is the first time that they have been used to comprehensively assess the impact of a diet on skin hydration and lubrication. Moreover, there are publications describing their repeatability and accuracy [62] and characterizing them in detail [63].

There are many methods for assessing the condition of skin, e.g., methods for assessing exposure to environmental pollution and oxidative stress, pigmentation, topography, radiance and color, elasticity, density, firmness and the structure of the microbiome [64]. These methods can be used to work with patients, as well as for research purposes. For example, there are reports suggesting that higher levels of the antioxidant substance lycopene in the skin correlate with lower roughness of skin [65].

In order for the epidermis to perform its function, it requires the presence of a hydrolipidic film on the surface and an appropriate degree of hydration to condition itself in response to external and internal factors. The effect of using dietary ingredients with antioxidant properties was assessed by Heinrich et al. [66]. The duration of the study was 12 weeks. The first group of volunteers (*n* = 13) received lycopene (3.0 mg), lutein (3.0 mg), beta-carotene (4.8 mg), alpha-tocopherol (10.0 mg) and selenium (75.0 mg); the second group (*n* = 13) received lycopene (6.0 mg), beta-carotene (4.8 mg), alpha-tocopherol (10.0 mg) and selenium (75.0 mg); the third group (*n* = 13) received a placebo. The above supplementations influenced the density and thickness of the epidermis. In the case of the density, the authors noted statistically significant changes of 6.57% (in the first group) and 7.01% (in the second group). The thickness analysis showed statistically significant changes of 15.99% in the first group and 14.09% in the second group.

Literature data show that both free radicals generated under the influence of UV radiation and those generated as a result of endogenous production can lead to photoaging of the skin [67], and in extreme cases, even to skin cancer [68]. Substances with antioxidant properties can be used externally or ingested. Local application of this type of substances allows them to be delivered directly to the skin surface in full without losses. Although the goal of the pharmaceutical and cosmetology industries is to increase the penetration of substances into the deeper layers of the skin, it is not high enough. On the other hand, oral ingestion of ingredients with antioxidant properties allows for reaching the deeper layers of the skin [69].

To sum up, our research shows that skin hydration was significantly influenced by the consumption of vitamins A and E, and skin lubrication was influenced by the consumption of Cu. Vitamin A and its derivatives contribute to the proper exfoliation of the stratum corneum, which improves its protective function and reduces transepidermal water loss [70]. Vitamin E, on the other hand, is one of the most effective antioxidants. By penetrating deep into the lipid barrier of skin cells, vitamin E seals and strengthens the cell membrane, which causes water retention [26]. The mechanism of the influence of Cu on skin lubrication has not, to our knowledge, been characterized in the literature. It is emphasized that Cu stimulates the proliferation of skin fibroblasts. It is a cofactor of superoxide dismutase and also prevents lipid peroxidation and oxidative damage to cell membranes [71].

The advantage of this study is the assessment of the actual intake of antioxidant ingredients with the standard diet of the study participants. Other studies, such as that by Dumoulin et al. [72], relied on participants taking the formulation developed by the authors. Our research allowed for assessing the effects of consumed antioxidants on skin hydration and lubrication; this can be the basis for rational supplementation that is tailored to the needs of patients. When the diet is rich in antioxidants, the use of additional dietary supplements seems unnecessary.

It should also be emphasized that the choice of products, diet balancing and eating habits of patients are influenced by a number of factors, such as social status; economic, cultural and adaptive factors; and interest in a healthy lifestyle [73,74,75,76]. Nutritional factors are one of the selected aspects because the proper hydration of the skin can also be influenced by genetic factors and not just a properly balanced diet.

Our study has several limitations. The analyses were carried out in a group of young women; it is worth also conducting them in a group of young men, but they show less willingness to participate in this type of research. The three-day dietary interview method of current recording is a very common method for assessing the consumption of selected ingredients, but it is subject to error due to the self-estimation of portion sizes by the participants. Research on the influence of factors on the level of skin hydration and lubrication may in the future also include a survey on skin care and the amount of water consumed by respondents, as well as assess the consumption of polyphenolic compounds.

Another aspect that needs to be investigated is the assessment of skin hydration and lubrication in other areas, such as the feet. This is especially important in diseases such as diabetes. Patients with this disease should take special care of their skin. The skin of the feet is characterized by the thickest layer of stratum corneum. It is dry and has a tendency toward crack formation. This favors the penetration of microorganisms, which results in infection [77].

Future research should focus on evaluating other factors that affect proper skin hydration and lubrication, as well as the correlation between the degree of skin hydration and lubrication and the risk of skin diseases.

## 5. Conclusions

Over one-third of students showed insufficient consumption of products rich in vitamins C, D and E, as well as zinc. The highest degree of hydration was observed in the areas of the eyelids, neckline and chin. However, the greatest amount of sebum was found in the areas of the nose and forehead. With increasing age, the level of cheek hydration decreased. A low positive correlation between hydration or lubrication and Cu, vitamin A and vitamin E was observed. In conclusion, to properly moisturize and lubricate the skin, young women should eat products that are rich in ingredients with antioxidant properties, in particular, fat-soluble vitamins A and E, but also copper.

## Figures and Tables

**Figure 1 antioxidants-10-01110-f001:**
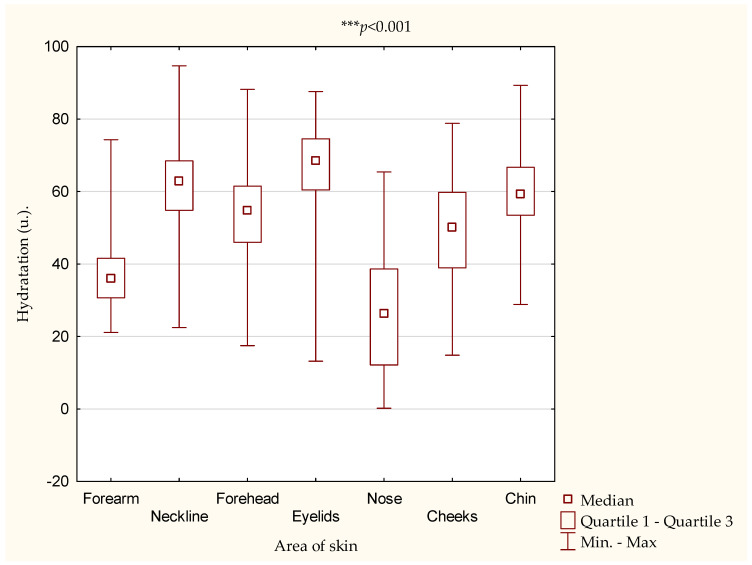
Differences in hydration of the body and face skin. Av.—average, SD—standard deviation.

**Figure 2 antioxidants-10-01110-f002:**
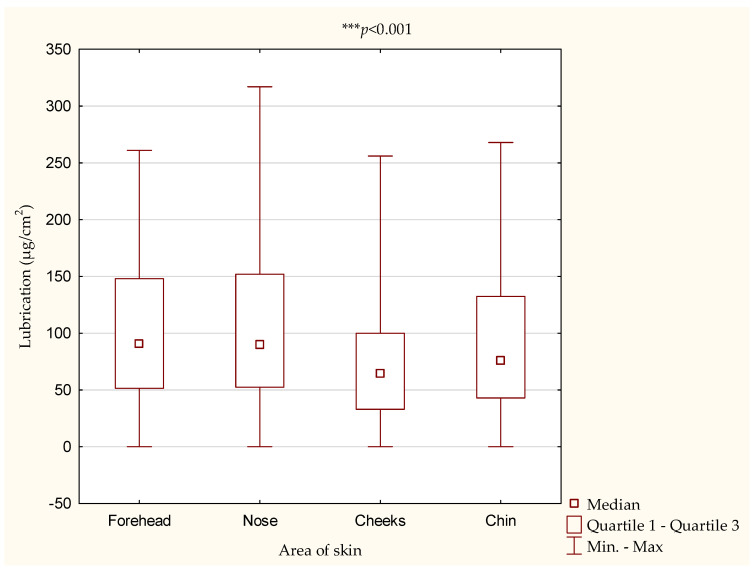
Differences in lubrication of the body and face skin. Av.—average, SD—standard deviation.

**Figure 3 antioxidants-10-01110-f003:**
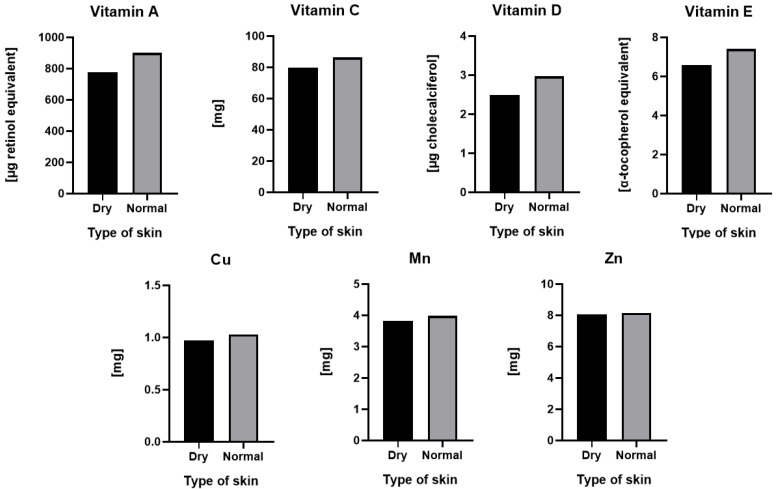
Consumption of antioxidant ingredients by young women with dry (*n* = 102) and normal (*n* = 69) skin: vitamin A, vitamin C, vitamin D, vitamin E, Cu, Mn and Zn.

**Table 1 antioxidants-10-01110-t001:** Characteristic of study group (*n* = 172).

Parameters	Av. ± SD	Min–Max	Med.	Q1–Q3
Age (years)	20 ± 1	18–25	20.00	21–21
Height (cm)	167 ± 6	155–182	168.00	163–171
Weight (kg)	62 ± 10	39–100	60.60	54–68
BMI (kg/m^2^)	21.9 ± 3.2	15.3–34.7	21.35	19.8–24.0

Av.—average, Max—maximum, Med.—median, Min—minimum, Q1—lower quartile, Q3—upper quartile, SD—standard deviation.

**Table 2 antioxidants-10-01110-t002:** Percentage of young women with sufficient and insufficient consumption of the tested antioxidant ingredients.

Component	Type of Norm	Norm (per Day)	Consumption Av. ± SD	Students with Insufficient Consumption (%)	Students with Sufficient Consumption (%)
Vitamin A (µg RE)	EAR	500	825.0 ± 688.1	27.3	-
Vitamin C (mg)	EAR	60	82.6 ± 54.1	39.5	-
Vitamin D (µg cholecalciferol)	AI	15	2.7 ± 2.9	-	1.2
Vitamin E (mg α-tocopherol equivalent)	AI	8	6.9 ± 4.1	-	26.7
Cu (mg)	EAR	0.7	1.0 ± 0.4	20.9	-
Mn (mg)	AI	1.8	3.9 ± 1.9	-	91.9
Zn (mg)	EAR	6.8	8.1 ± 2.3	32.0	-

AI—adequate intake, Av.—average, EAR—estimated average requirement, RE—retinol equivalent, SD—standard deviation.

**Table 3 antioxidants-10-01110-t003:** Descriptive characteristics of the body composition parameters estimated by electrical bioimpedance method.

Parameter	Av. ± SD	Min–Max	Med.	Q1–Q3
Total body water mass (kg)	32.40 ± 3.75	23.40–44.90	32.25	29.55–35.05
Extracellular water mass (L)	12.32 ± 1.44	9.00–17.40	12.20	11.20–13.20
Intracellular water mass (L)	20.09 ± 2.33	14.40–27.50	20.05	18.30–21.75
ECW/TBW	0.380 ± 0.005	0.363–0.392	0.380	0.377–0.384
ECF/TBF	0.333 ± 0.004	0.317–0.345	0.334	0.330–0.337
Fat free mass (kg)	44.27 ± 5.15	31.80–61.20	44.05	40.30–47.90
Skeletal lean mass (kg)	41.62 ± 4.82	30.00–57.50	41.40	37.90–45.10
Skeletal muscle mass (kg)	24.20 ± 3.04	16.77–33.90	24.13	21.90–26.35
Protein mass (kg)	8.68 ± 1.01	6.20–11.90	8.70	7.90–9.40
Body fat mass (kg)	17.34 ± 6.65	5.40–44.50	16.45	12.15–21.15
Percent body fat (%)	27.39 ± 6.47	13.90–45.50	27.56	22.40–31.40
Mineral mass (kg)	3.18 ± 0.39	2.19–4.39	3.14	2.89–3.45
Bone mineral content (kg)	2.65 ± 0.34	1.84–3.66	2.62	2.42–2.86
Basal metabolic rate (kcal)	1326.19 ± 111.18	1057.71–1240.70	1321.35	1240.70–1405.05
Visceral fat area (cm^2^)	58.89 ± 23.13	10.20–142.20	56.08	42.95–69.38
Abdomen circumference (cm)	78.89 ± 8.83	58.30–111.30	77.45	72.65–84.30
Hip circumference (cm)	93.62 ± 5.61	80.10–114.7	92.95	89.65–96.95
Chest circumference (cm)	88.29 ± 5.93	73.90–110.00	87.55	84.10–92.10
Arm muscle circumference (cm)	22.57 ± 1.58	18.85–27.77	22.57	21.43–23.60
Fitness score (points)	72.77 ± 5.08	56.00–84.00	73.00	70.00–76.00

Av.—average, ECF—extracellular fluid, ECW—extracellular water, Max—maximum, Min—minimum, Q1—lower quartile, Q3—upper quartile, SD—standard deviation, TBF—total body fluid, TBW—total body water.

**Table 4 antioxidants-10-01110-t004:** Percentage of young women with each degree of hydration of the examined skin areas.

Area	Women with Skin Type (%)		
	Very Dry	Dry	Sufficiently Moisturized
Cheeks	7.6	18.0	74.4
Chin	0.6	3.4	96.0
Eyelids	1.2	0.6	98.2
Forearm	23.3	45.3	31.4
Forehead	3.5	8.1	88.4
Neckline	1.2	1.2	97.6
Nose	59.3	19.2	21.5

**Table 5 antioxidants-10-01110-t005:** Percentage of young women with each degree of lubrication of the examined skin areas.

Area	Women with Skin Type (%)		
	Dry	Normal	Oily
Cheeks	53.5	39.5	7.0
Chin	64.5	33.2	2.3
Forehead	55.8	37.8	6.4
Nose	57.6	33.7	8.7

**Table 6 antioxidants-10-01110-t006:** Correlations between skin hydration, lubrication and body composition (*p* < 0.05).

Factor 1	Factor 2	*r*
Body moisturizing	ECW/TBW	−0.15
Moisturizing the cheeks	Age	−0.19
Total lubrication	ECF/TBF	0.15

ECF—extracellular fluid, ECW—extracellular water, TBF—total body fluid, TBW—total body water.

**Table 7 antioxidants-10-01110-t007:** Correlations between skin hydration, lubrication and intake of antioxidant components (*p* < 0.05).

Factor 1	Factor 2	*r*
Vitamin A	Hydration of the neckline	0.16
Vitamin A	Hydration of the forearm	0.16
Vitamin A	Hydration of the face	0.18
Vitamin A	Total hydration	0.17
Vitamin E	Hydration of the neckline	0.17
Vitamin E	Hydration of the face	0.16
Vitamin E	Total lubrication	0.15
Cu	Hydration of the face	0.16
Cu	Total lubrication	0.17

## Data Availability

Data is contained within the article and Appendix A.

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
