# Peer review of "Intake of Antioxidant Vitamins and Minerals in Relation to Body Composition, Skin Hydration and Lubrication in Young Women"

_antioxidants, 2021, doi:10.3390/antiox10071110_

Round 1

Reviewer 1 Report

The authors identified the skin health improvement effect of antioxidants through meta-study. Validation through meta-study is an important process. However, Textbook-like content does not attract the reader's attention.

Major comments

  1. It is difficult to understand because many contents are simply presented. Papers should be logical.
  2. Unify the form of the figure and redraw it according to the paper. It would be better to use a graphing program other than Excel.
  3. The authors should be more specific about the conclusions of this paper. Clearly articulate the purpose of the paper.

Reviewer 2 Report

All my suggestions has been addressed

Reviewer 3 Report

Manuscript Number: antioxidants-1269588 titled: Intake Of Antioxidant Vitamins And Minerals In Relation To Body Composition, Skin Hydration And Lubrication In Young Women

Review 1 – 21 June 2021

Dear Editor of Antioxidants

To the Editor (in general)

 Dear Editor, the argument is interesting but the Introduction has to be completed and the beneficial effects of vitamins have to be better treated. The M&M section has to be more detailed. The discussion of data has to be completed with comments on some reference more. The Conclusions have to be improved, detailed, extended (four lines more are not enough).

I suggest a major revision.

To the Authors (in detail)

  • In the title: of, and, to, in (in small letters);
  • Abstract section. The abstract could be the only part of the manuscript included in some databank. Please insert some relevant result more at the end of this section;

  • Introduction section, line 43, support better your statement and insert 1-2 reference more in addition to your ref.4;
  • Introduction section, lines 60-62, extend this statement with other beneficial effects of Vit. C and insert 1-2 reference more;
  • Introduction section, lines 64-68, extend this statement with other beneficial effects of Vit. D and insert 1-2 reference more;
  • Introduction section, lines 69-73, extend this statement with other beneficial effects of Vit. E and insert some reference more;

  • 3.0 section please, move to 2.1 sub-section, the range of age, height, weight (lines 169-170) and table 1, these are a part of the M&M section and not a result;
  • 1 section, detail from how many towns/cities the group was selected. In addition, how many smokers? How many years have they been smoking?

  • Page 7, Line 215, in your table chin is 95.6 and not 95.9. Please, compare all data reported in the discussion with data listed in the tables;
  • Page 12, lines 301-304. Please, briefly discuss here the fruits and vegetables in which Vitamin A content can be found. I have suggested one reference, but you have to add one reference more. Here, find, read and discuss:

-Temperature and storage time increase provitamin A carotenoid concentrations and bioaccessibility in post-harvest carrots

Food Chemistry 338 (2021) 128004. https://doi.org/10.1016/j.foodchem.2020.128004

  • Page 12, lines 312-316. Please, briefly discuss here the fruits and vegetables in which the highest Vitamin C content can be found. For example: citrus juice and apple juice. Support this statement with some reference, find read and discuss:

- Bergamot (Citrus bergamia, Risso): The Effects of cultivar and harvest date on functional properties of juice and cloudy juice.

Antioxidants 8, 221 (2019). doi:10.3390/antiox8070221

-Ascorbic acid content in apple pulp, peel, and monovarietal cloudy juices of 64 different cultivars

INTERNATIONAL JOURNAL OF FOOD PROPERTIES 2017, VOL. 20, NO. S3, S2626–S2634. https://doi.org/10.1080/10942912.2017.1381705

  • Page 12, lines 325-327. Please, briefly discuss here the fruits and vegetables in which Vitamin E content can be found. I have suggested one reference, but you have to add one reference more. Here, find, read and discuss:

-Phenolic and fatty acid profiles, α-tocopherol and sucrose contents, and antioxidant capacities of understudied Portuguese almond cultivars.

Journal of Food Biochemistry, Volume 43, Issue 7July 2019 Article number e12887

            DOI: 10.1111/jfbc.12887

  • Page 12, when you discuss about Vitamin, please, briefly discuss here the foods in which Vitamin D can be found and insert 2 references to support your statement

  • Line 285: one person;

  • Line 298, verify the hyphen;

  • Page 12, please, improve the discussion about the quantity of each vitamin suggested per day (or week/month) per person, by an International organization for human health and compare these suggested data with the intake of your studied population. Please, use references easy to be find by each reader (no thesis, no proceedings of regional congress, no regional journals, no regional regulations);

  • Lines 310, 315, 326, 340, 397, 428 and in the whole manuscript, include the bibliography as suggested by Antioxidants, do not include the year of publication;
  • Tables and figures have to be better discussed. For example, table 7 is almost non-discussed;
  • Conclusions non conclusions (extend, improve, argue);
  • In the bibliography you have included many references in Polish, but these are difficult or impossible to be read by the International readers. Please, do not replace them but add other references in English;
  • Very important, please, write in blue color or evidence the corrections you will do;

     Regards.

Round 2

Reviewer 1 Report

Everything suggested by the reviewer was well revised.

Reviewer 3 Report

Manuscript Number: antioxidants-1269588 titled: Intake of Antioxidant Vitamins ond Minerals in Relation to Body Composition, Skin Hydration and Lubrication in Young Women

Review 2 – 4 July 2021

Dear Editor of Antioxidants

 the argument is interesting, the experiment is well designed and data are well discussed. The authors have included all comments I have listed.

There is some correction more to do.

I suggest a minor revision.

  • Line 81, verify the font size;
  • Line 152, separate 3.0 from mg;
  • Line 417, verify the font size;
  • References section, ref.15: Citrus bergamia in italic. Please, verify the whole section and write the scientific names in italic.

Regards.

Author Response

This manuscript is a resubmission of an earlier submission. The following is a list of the peer review reports and author responses from that submission.

Round 1

Reviewer 1 Report

The authors revealed the correlation between antioxidant intake and skin hydration.

Major comments

  1. The authors are verifying through clinical trials that very common-sense antioxidants are good for skin health. If this is the first experiment, it is a very meaningful experiment. However, there are many more controlled experiments. The authors should clarify what advantages this study has and what it is meaningful for, unlike previous studies. Clin Cosmet Investig Dermatol.2016; 9: 315–324 etc

  1. When a lot of antioxidant foods are consumed, various factors can be involved, such as social status, wealth, and interest in health. The reason why the subject's skin moisturization is good may not be because he consumes antioxidant foods, but because he has good skin and is able to consume a lot of antioxidant foods because he is rich. Relatively poor people may not consume antioxidant foods and may have poor skin moisturization. There is a need to provide specific explanations for these uncontrolled variables and disccusion on existing studies.

Minor comments

  1. The authors would like to rewrite the figure. Draw a graph in the form of a scientific paper

Reviewer 2 Report

Dear authors, this is a very interesting research and article and I have no important concerns about it. However, it would be desirable a little bit more information about the level of hydration in a frequently dry skin as its the foot. It could de placed in the discussion section. 

Reviewer 3 Report

Although there are limitations in clinical trials, the intake of antioxidant-related vitamins and skin characteristics were evaluated during routine dietary intake. Among the nutrients consumed through food, not only vitamins, but also polyphenols derived from ingestion of fruits and vegetables impart antioxidant activity, so it is too unreasonable to interpret this as a result of only vitamins excluding them.